# Biofunctionalized Decellularized Tissue-Engineered Heart Valve with Mesoporous Silica Nanoparticles for Controlled Release of VEGF and RunX2-siRNA against Calcification

**DOI:** 10.3390/bioengineering10070859

**Published:** 2023-07-20

**Authors:** Wenpeng Yu, Xiaowei Zhu, Jichun Liu, Jianliang Zhou

**Affiliations:** 1Department of Cardiovascular Surgery, The Second Affiliated Hospital of Nanchang University, No. 1 Minde Road, Nanchang 330006, China; ywp19940515@163.com; 2Department of Cardiology, Zhongshan Hospital, Fudan University, Shanghai 200032, China; 17111210037@fudan.edu.cn

**Keywords:** decellularized valve, mesoporous silica nanoparticle, RunX2-siRNA, VEGF, calcification

## Abstract

The goal of tissue-engineered heart valves (TEHV) is to replace normal heart valves and overcome the shortcomings of heart valve replacement commonly used in clinical practice. However, calcification of TEHV is the major bottleneck to break for both clinical workers and researchers. Endothelialization of TEHV plays a crucial role in delaying valve calcification by reducing platelet adhesion and covering the calcified spots. In the present study, we loaded RunX2-siRNA and VEGF into mesoporous silica nanoparticles and investigated the properties of anti-calcification and endothelialization in vitro. Then, the mesoporous silica nanoparticle was immobilized on the decellularized porcine aortic valve (DPAV) by layer self-assembly and investigated the anti-calcification and endothelialization. Our results demonstrated that the mesoporous silica nanoparticles delivery vehicle demonstrated good biocompatibility, and a stable release of RunX2-siRNA and VEGF. The hybrid decellularized valve exhibited a low hemolysis rate and promoted endothelial cell proliferation and adhesion while silencing RunX2 gene expression in valve interstitial cells, and the hybrid decellularized valve showed good mechanical properties. Finally, the in vivo experiment showed that the mesoporous silica nanoparticles delivery vehicle could enhance the endothelialization of the hybrid valve. In summary, we constructed a delivery system based on mesoporous silica to biofunctionalized TEHV scaffold for endothelialization and anti-calcification.

## 1. Introduction

Valvular heart disease is a common and severe progressive disease, with about 40% of individuals >70 years old having mild calcification of the aortic valve and >10% having severe calcification [1]. Valvular heart disease is associated with a high risk of cardiovascular events even when no blood flow is present, and the mechanics significantly block the left ventricular outflow [2]. Valve replacement is the main option to improve clinical outcomes and increase the survival rate if patients cannot effectively intervene in the course of the disease [3]. However, every valve replacement device has a unique problem: mechanical valves cause emboli in the body, which must be treated for a lifetime post-surgery and lead to secondary anticoagulant-related bleeding [4]. On the other hand, the heart valve consists of homologous or xenografted tissue from humans or animals, usually pigs or cattle, and has been previously treated with various physical chemistry processes to improve its performance [5]. The major disadvantage of these valve replacements is calcification upon prolonged exposure to circulating blood [6]. Valve calcification has always been a challenge to researchers and clinicians.

The concept of TEHV is described as the development of a heart valve with mechanical function and bioactivity under physiological conditions that facilitate the repair and reshaping of the scaffold [7]. TEHV addresses the limitations of the implant by combining the biomaterials with autologous cells for growth and biointegration. Ideal cardiac valve prostheses have antithrombotic, biocompatibility, durability, anti-calcification, and physiological hemodynamic characteristics [8]. However, valve calcification has been a major cause of failure in biological valves and TEHV [9]. Decellularized matrixes exhibit good biocompatibility and provide a microenvironment for cell adhesion, proliferation, and differentiation, and hence, have been widely applied in tissue engineering [10]. However, the surface of the decellularized matrix is not covered by endothelial cells, which would lead to a constant infiltration of plasma components, which are likely to be a very important cause of tissue calcification [11,12]. Based on this problem, the ability of decellularized materials to achieve endothelialization is crucial. Imagine if, while endogenously achieving anticalcification, a substance could synergize with anticalcification and locally aggregate more endothelial cells, which might greatly improve the current problem. VEGF is the most effective pro-endothelialization factor and a key regulator of endothelialization that can synergistically enhance tissue regeneration [13]. Based on our previous research, we tried to load VEGF into the DPAV through different modification methods, such as Polyethylene glycol (PEG)-mediated covalent binding of VEGF to decellularized aortic valves [14], and covalently immobilize the VEGF-loaded Polycaprolactone (PCL) nanoparticles on the DPAV through the Michael addition reaction [15], thereby exerting its endothelial effect. There have also been studies using polyelectrolyte multilayer membrane self-assembly to immobilize VEGF on the DPAV to reduce platelet adhesion [16].

The advent of nanotechnology has made it possible to fight cancer and provide new insights into the TEHV. The mesoporous silica nanoparticle (MSN) has a nanoporous channel structure, a uniform and adjustable pore size, and a dominant-specific surface area, with application prospects in biocatalysis and the assembly of functional polymer complexes [17]. Mesoporous silica can bind with various functional groups of active pharmaceutical ingredients for targeted delivery to the desired site of action. These dominant features make the MSN a promising nanoparticle carrier, revolutionizing different drug delivery methods [18] such as controlled drug delivery [19], targeted drug delivery [20,21], sustained drug delivery [22], and response systems [23,24].

While many studies have used nanocarriers to deliver siRNAs or DNA where they want to deliver, our study aims to address anti-calcification while allowing more endothelial cells to aggregate and tissues to rebuild faster. There is no research on using mesoporous silica to assemble VEGF and siRNA and apply it to valve tissue engineering. The aim of this study was to prepare a DPAV capable of delivering siRNA and VEGF to achieve the endogenous anti-calcification of siRNA, the exogenous synergistic anti-calcification of VEGF, and promote the rapid reconstruction of valve implants. Based on the characteristics of mesoporous silica, VEGF was encapsulated into the pore, and siRNA was complexed by electrostatic adsorption to prepare the MSN@VEGF-PEI-siRNA delivery system (the synthetic procedure for the MSN@VEGF-PEI-siRNA delivery system is summarized in Figure 1). In addition, the mesoporous silica nano-delivery system was complexed to a decellularized valve through heparinization and applied to a tissue engineering hybrid valve. In addition, the anti-calcification and endothelialization capabilities of the hybrid valve were explored.

## 2. Materials and Methods

Tetraethyl orthosilicate (TEOS), branched polyethyleneimine (PEI, 25 kDa), N-hydroxysuccinimide (NHS), cetyltrimethylammonium chloride solution (CTAC), triethylamine (TEA), and dimethyl sulfoxide (DMSO) were purchased from Sinopharm Chemical Reagent (Shanghai, China). VEGF was obtained from PeproTech (Cranbury, NJ, USA). DMEM, trypsin, type Ⅱ collagenase, penicillin/streptomycin, fetal bovine serum, and Alizarin red were purchased from Sangon Biotech (Shanghai, China). Porcine aortic valves were purchased from Songlin Meat Food Co., Ltd. A Cell Counting Kit-8 (CCK-8) and a BCIP/NBT ALP staining kit (Beyotime, Shanghai, China). An alkaline phosphatase assay kit (Jiancheng Bioengineering Institute, Nanjing, China), a calcium content detection kit (MingDian, Shanghai, China), and an RNeasy mini kit were purchased from Qiagen (Duesseldorf, Germany), IL-1, IL-6, IL-10, and TNF-α kit (R&D, Santa Clara, CA, USA). Primary antibodies (GAPDH, OPN, OSX, RunX2, α-SAM, Vimentin, and CD31) and donkey anti-rabbit IgG were procured from Abcam (Cambridge, UK). LipofectamineTM 3000 Transfection Reagent was purchased from Invitrogen (Carlsbad, CA, USA).

### 2.1. Tissue Processing, Cell Culture, and Calcification Model In Vitro

The rats were soaked in anhydrous ethanol for 5 min. The heart was removed, placed in phosphate-buffered saline (PBS) buffer containing 1% double-antibody, and the blood was washed. The disinfected minimally invasive surgical scissors were used to cut the heart along the route of the apical and aortic outflow tract, following which the mitral and aortic valves were exposed and cut off. The valve leaflets were placed into 6-well plates after PBS flushing, followed by the addition of type Ⅱ collagenase (2 mg/mL) and incubated at 37 °C and 5% CO_2_ for 3 h. The mixture was blown, resuspended, and transferred to a conventional medium for culture.

Valvular interstitial cells (VICs) were maintained in DMEM supplemented with 20% fetal bovine serum and 1% penicillin-streptomycin. ECs (purchased from the ATCC, USA) and Macrophages (RMa-bm)were maintained in DMEM supplemented with 10% fetal bovine serum and 1% penicillin-streptomycin. Cells were cultured at 37 °C with 5% CO_2_.

**Identification of VICs:** Cells were removed after fusion, rinsed three times with PBS, fixed with 4% paraformaldehyde for 30 min, and permeabilized with 0.2% Triton-X 100 for 2–5 min. Then, the cells were blocked with 5% BSA at room temperature for 30 min, followed by incubation with primary antibodies (α-SAM and Vimentin) in a humid chamber at 4 °C overnight and a secondary antibody in the dark for 30 min. After PBS washes, the cells were observed under a fluorescence microscope.

OIM (sodium dihydrogen phosphate 10 mmol/L, L-ascorbic acid 10 μg/mL, and dexamethasone 10 nmol/L were added into the complete medium) was used to culture VICs and promote their osteogenic differentiation. VICs cultured in a complete medium comprised the control group (DM). VICs were identified by immunofluorescence, and calcification-related genes (RunX2, OPN, OSX, BSP, OCN, and IL-6) were detected by RT-qPCR. Alizarin red staining, ALP staining, and calcium content were detected to evaluate the model.

### 2.2. RunX2-siRNA against Valvular Interstitial Cells Calcification

RunX2-siRNA was obtained from Sangon Biotech Co., Ltd. (Shanghai, China). siRNA-RunX2-1667 (5′–3′, sense: GCACGCUAUUAAAUCCAAATT, antisense: UUUGGAUUUAAUAGCGUGCTT), siRNA-RunX2-1550 (5′–3′, sense: GCACUCCAUAUCUCUACUATT, antisense: UAGUAGAGAUAUGGAGUGCTT), and RunX2-827 (5′–3′, sense: CCAUAACGGUCUUCACAAATT, antisense: UUUGUGAAGACCGUUAUGGTT) were synthesized. The anti-calcification capability of Runx2-siRNA was evaluated by RT-PCR, Western blot, Alizarin red staining, and ALP staining.

**PCR analysis:** VICs were treated with each group of samples for one week, cells were collected, and total RNA was extracted using the RNeasy Mini kit. After reverse transcription, real-time PCR was performed to detect the expression of the three main calcification genes RunX2, OPN, and osterix by RT-qPCR using GAPDH as the internal reference gene. The PCR primers are listed in Appendix A.

**ALP activity assay and staining:** ALP staining is an indicator of early osteogenic differentiation. Five days after the VIC culture, we used the BCIP/NBT ALP staining kit for ALP staining, according to the manufacturer’s instructions. The activity of ALP was directly measured using the alkaline phosphatase assay kit.

**Alizarin red staining and determination of calcium content:** Briefly, after treating VIC with each group of samples for 9 days, the medium was removed and washed three times with PBS, and the cells were fixed in 4% paraformaldehyde for 10 min, followed by three washes with PBS. Subsequently, 1% alizarin red dye was added to the cells, the amount added was sufficient to cover the cells, and the staining was washed well after 3 min and observed under the microscope. For quantitative analysis of the alizarin red stain, 1 mL of 10% CPC was added to each well of the 6-well plate. Then, it was gently shaken on a shaker for 20 min and transferred to the eluted CPC into an EP tube. The eluted CPC was diluted 20 times with 10% CPC and measured at a wavelength of 562 nm. Finally, the relative calcium content of the total protein was directly determined using a calcium content assay kit. Briefly, remove the culture medium and wash once with PBS solution. Add 100 µL of sample lysis buffer to fully lyse the cells. After thorough lysis, centrifuge at 4 °C for 10,000× *g* for 5 min and collect the supernatant. Measure the absorbance at 575 nm.

### 2.3. Preparation of MSN@VEGF-PEI-siRNA Delivery System

MSN was synthesized according to the previously published protocols with minor modifications [25]. Briefly, 2.0 g of the CTAC was dissolved in 20 mL deionized water. Then, 50 μL of TEA was added, and the mixture was stirred at 95 °C for 1 h. Then 1.5 mL TEOS was added dropwise, and the solution was stirred for another 1 h. The products were collected by centrifugation and refluxed in HCl/methanol for 6 h to remove the template CTAC. Finally, the MSN was collected by centrifugation and vacuum drying.


**Grafting of carboxylic groups at the surface of MSN by salinization (MSN-COOH):**


The grafting of carboxylic groups on the surface of MSN was synthesized. According to a previous study [26], 150 mg of MSN was degassed under a vacuum for 12 h and suspended in 15 mL of toluene. Then, 34 μL of TESP (120 μmol) was added dropwise to the mixture. After 24 h of stirring under a static N_2_ atmosphere, the obtained solid was washed three times with a mixture of diethyl ether and DCM (1:1, *v*/*v*) and dried at 60 °C for 12 h in air. Subsequently, 600 mg of MSN was refluxed for 1 h in a mixture of NH_4_NO_3_/EtOH (2 g/100 mL). This procedure was repeated four times, and centrifugation (12,000× *g*, 10 min, 20 °C) was used to wash the particles between each step. After the complete removal of the solvent under vacuum, the powder was stored in a desiccator. These nanoparticles were referred to as MSN-COOH.

**VEGF was loaded into the MSN-COOH:** To load VEGF into MSN-COOH, MSN-COOH (20 mg) was dispersed in PBS (4 mL) containing VEGF for 6 h. Excessive VEGF was removed by centrifugation and repeatedly washed with PBS (pH 7.4). The resulting particles are represented as MSN@VEGF-COOH.

**Synthesis of MSN@VEGF-PEI-siRNA:** An equivalent of 100 mg of NHS was solubilized in DMSO before adding 5 mL of ultrapure water, EDC (160 mg dissolved in 5 mL of ultrapure water), and 10 mL of MSN@VEGF-COOH suspension (150 mg in 10 mL of water, pH = 7). PEI (450 mg of a solution at 50% in weight) was solubilized in 5 mL of water, added to the reaction mixture, and stirred at 20 °C for 24 h. The resulting particles are represented as MSN@VEGF-PEI. RunX2-siRNA was incubated in the MSN@VEGF-PEI suspension at room temperature for 30 min to form the MSN@VEGF-PEI-siRNA delivery system by electrostatic absorption. The samples were characterized by scanning electron microscope (SEM, JSM-6700F, JEOL) and transmission electron microscope (TEM, HT7700, Hitachi).

### 2.4. Encapsulation Efficiency and In Vitro Release Study

The encapsulation efficiency was calculated as follows: encapsulation efficiency (%, *w*/*w*) = (feed drug content − free drug content)/feed drug content × 100%.

MSN@VEGF-PEI-siRNA (2 mg) was dispersed in PBS (5 mL). The mixture was agitated in a constant temperature shaker. At various time points of 1 h, 2 h, 4 h, 8 h, 12 h, 1 d, 2 d, 4 d, and 7 d, the supernatant was withdrawn and replaced with fresh buffer. The release of VEGF from the pore was monitored by measuring the absorbance at 450 nm. RunX2-siRNA content in the supernatant was determined by monitoring the 260 nm absorbance band. Finally, the cumulative release percentage of VEGF and RunX2-siRNA was calculated.

### 2.5. Cytotoxicity Assay

VICs were seeded in 96-well plates at a density of 1 × 10^4^ cells/well. After the cells were fully adhered to the wall, the cells were treated with each set of samples for 24 h. An appropriate amount of MTT solution was added to each well and the absorbance at a wavelength of 570 nm was measured.

### 2.6. Cellular Uptake Study

The cellular uptake of MSN@VEGF-PEI-siRNA was confirmed by fluorescence microscopy. VICs were seeded in 24-well plates at a density of 1 × 10^4^ viable cells/well and incubated overnight to allow cell attachment. The medium was replaced with a fresh medium containing MSN@VEGF -PEI-siRNA(-FAM). After incubation for 4 h, the VICs were washed three times, fixed with 4% paraformaldehyde for 20 min, and observed under the microscope.

### 2.7. Inflammatory Factor Detection

Macrophages were cultured in six-well plates, and each group of samples was added when the cells were fully apposed. Then, 24 h later, the macrophage supernatant was collected, and IL-1, IL-6, IL-10 and TNF-a were detected using ELISA kits according to the procedure.

### 2.8. Efficiency of the MSN@VEGF-PEI-siRNA Delivery System for Gene Silencing in VICs

The experiment was divided into four groups (MSN@VEGF-PEI-siRNA, Lipo 3000+siRNA, MSN, and Con). We evaluated the anti-calcification capability of RunX2-siRNA by RT-qPCR, Alizarin red staining, ALP staining, and calcium content.

### 2.9. Endothelialization Rate of the MSN@VEGF -PEI-siRNA Delivery System

**ECs adhesion and proliferation assay:** Cells were seeded at a density of 2 × 10^4^ cells in each well of a 96-well plate, and samples (MSN@VEGF-PEI-siRNA, MSN-PEI-siRNA, MSN, and Con; the control group was marked as “Con”) were added to the plate. After 2, 4, or 12 h, the wells were washed with PBS three times. Then, 10 μL of CCK-8 solution was added to each well, and the absorbance was measured at 450 nm. Cells were seeded into a 96-well plate at a density of 2 × 10^4^ cells in order to study the proliferation of ECs across different groups. Following this, 10 µL of CCK-8 reagent was added to each well after 1, 3, or 5 days of incubation. After a further 2-h incubation, the absorbance at 450 nm was measured on a microplate reader. 

**ECs’ angiogenic determination:** ECs were seeded in 6-well plates and when the cell density reached 100%, the cells were scored straight with the pipette tip and the scraped cells were thoroughly washed. Samples were then added for processing and microscopic images were taken at two time points, 0 and 6 h after scratching. ImageJ software was used to quantify the images.

**Tube formation assay:** All consumables used in the experiment need to be pre-cooled in advance. Then, 50 µL of Matrigel was placed into 96-well plates using the pre-cooled tips and incubated in a cell incubator for 30 min. Samples were then added to the well plates and ECs were seeded at a density of 4 × 10^5^ cells/well and incubated in the incubator for 6 h before images were taken under the microscope. Images were quantitatively analyzed using ImageJ software.

### 2.10. Preparation and Characterization of the DPAV Modified with MSN@VEGF -PEI-siRNA

**Preparation of DPAV:** Decellularization was performed as previously described [27]. Briefly, the porcine aortic valves were exposed to a solution consisting of 1% poly (ethylene glycol)–poly (3-caprolactone) and were continuously agitated for 24 h at 37 °C; 75 rotations per minute on an incubator shaker. After that, the decellularized porcine aortic valve was treated with a PBS solution containing 0.2 mg/mL DNase I and 20 mg/mL RNase A, and was kept on an incubator shaker at 37 °C; 75 rotations per minute for 2 h. The resulting product was stored at −80 °C for later use.

**Preparation of modified hybrid valve (DPAV-MSN@VEGF-PEI-siRNA):** Polyelectrolyte multilayer film is composed of polyelectrolytes with opposite charge alternately adsorbed on the surface of the material through electrostatic interaction. Heparin (HEP) itself contains a large amount of negative charge, and the surface heparinization of the decellularized valve makes its surface heparin an anionic polyelectrolyte. The polyethyleneimine (PEI) in MSN@VEGF-PEI-siRNA is a cationic polyelectrolyte, and the polyelectrolyte layer technology was used for the pre-heparinization of the valve bind to the heparinized decellularized valve through electrostatic effect and MSN@VEGF-PEI-siRNA with a positive charge. Briefly, the prepared valve was stored in PBS at 4 °C, and the surface structure was observed by SEM. It should be noted that the synthesis steps for DPAV-MSN-PEI-siRNA and DPAV-MSN@VEGF-PEI are the same as those described above, while MSN-PEI-siRNA is a nanoparticle without VEGF loading and only containing siRNA, and MSN@VEGF-PEI is the opposite.

### 2.11. In Vitro Release Assay of MSN@VEGF-PEI-siRNA Nanocomposite-Modified DPAV

The MSN@VEGF-PEI-siRNA-modified decellularized valve was placed in a liquid with appropriate release media (such as buffer solution and serum) and then continuously oscillated in a thermostatic oscillator at 37 °C. The samples were withdrawn at different time points, and the concentration of MSN@VEGF-PEI-siRNA was determined on a spectrophotometer. After each sampling, an equivalent volume of release media was replenished. The cumulative release percentage was calculated.

### 2.12. Hemolysis Tests

Hemolysis tests were performed on each valve. Briefly, the processed valves (DPAV-MSN@VEGF-PEI-siRNA, DPAV-MSN@VEGF-PEI, DPAV, and PAV, n = 3) (1 cm × 1 cm) were placed in centrifuge tubes containing 10 milliliters of Phosphate Buffered Saline (PBS) and allowed to settle at 37 degrees Celsius for 1 h. Subsequently, 0.2 milliliters of diluted rabbit blood (diluted with PBS) was added to each centrifuge tube and incubated for another hour. Tri-distilled water was employed as the positive control, while PBS served as the negative control. The supernatant was collected by centrifugation at 1500 rpm for 10 min, and the absorbance was measured at 545 nm. The HR was calculated using the following formula: HR = (V − N)/(P − N), where V, N, and P represent the absorbance values of the supernatant in the valve group, the negative control group, and the positive control group, respectively.

### 2.13. Cell Seeding

VICs and rat endothelial cells (ECs) were planted on each valve. VICs were cultured in a calcified medium after implantation. Genes associated with osteogenic differentiation were detected by RT-qPCR, and calcium content was measured. Cell proliferation was detected with the CCK-8 kit

### 2.14. Biomechanical Characterization of MSN@VEGF-PEI-siRNA Nanocomposite-Modified DPAV

We performed uniaxial tensile loading tests to delineate the biological mechanical properties of each valve. Considering the valve exhibits direction anisotropy, we employed the Computer Servo Control Tensile Tester to acquire pertinent data on the related biological and mechanical properties of the circumferential and radial directions. Following Zhu and colleagues’ method [3], 15 mm × 5 mm and 10 mm × 5 mm strips were cut from the circumference and radial direction of each valve, respectively. These were then fixed between clamps on the tester before a constant speed of 10 mm/min was applied until the samples failed. Load-displacement data from the uniaxial tensile loading tests were used to generate stress-strain curves. The elastic modulus, ultimate tensile strength, fracture strength, and fracture strain were calculated from these curves. The length, width, and thickness of the valves were measured using digital calipers, with all tests performed at room temperature.

### 2.15. CD31 Immunofluorescence Staining

To assess the in vivo biocompatibility and regeneration capability of the modified hybrid valve, we subcutaneously implanted DPAV-MSN@VEGF-PEI-siRNA, DPAV-MSN@VEGF-PEI, DPAV, and PAV in SD rats. The Institutional Animal Care and Use Committee at Zhongnan Hospital of Wuhan University approved the proposed protocols (approval code: ZN2023013). After disinfecting and shaving the skin, valves measuring 1 cm × 1 cm were subcutaneously implanted in rats along both sides of their backs. The incisions were then closed using skin sutures. Each rat had one valve implanted on each side. Following the surgery, rats were allowed to be reared under normal conditions. After one month, the rats were euthanized, and the valves were extracted from the subcutaneous tissue of their backs. The valves were then stained using anti-CD31 antibody in accordance with immunofluorescence staining procedures.

### 2.16. Statistical Analysis

The data were presented as mean ± standard deviation. One-way analysis of variance (ANOVA) with the Student-Newman-Keuls method was performed for the statistical analysis of intergroup measurement variability. A *p*-value of less than 0.05 was considered statistically significant.

## 3. Results and Discussion

### 3.1. Anti-Calcification Research of RunX2-siRNA

The data of the valvular interstitial cells (VICs) calcification model are shown in Appendix A. We synthesized three small interfering fragments and screened them to identify the RunX2-siRNA with the highest silencing efficiency. As shown in Appendix A, the mRNA expression of RunX2, OPN, and OSX was decreased in three different interfering fragments after the silencing of RunX2, and the silencing efficiency of RunX2-siRNA-827 was the highest. Subsequently, we collected cell samples and evaluated calcified nodules by Alizarin red and ALP staining (Appendix A). The number of calcified nodules in the transfected group is shown in Appendix A, and was significantly lower than that in the Con and NC groups. The results of Alizarin red staining and quantitative analysis were consistent with the above findings. Hence, RunX2-siRNA-827 was chosen for the next step.

According to the pathological characteristics, the calcified valve has the characteristic of bone tissue and VICs eventually cause calcium deposition due to the regulation of certain upstream genes by the formation of calcium phosphate solids/precipitate. Some studies have shown that valve cells have osteogenic differentiation during valve calcification [28]. Calcified valve highly expressed early osteogenesis-related gene RunX2 [29]. RunX2 is known as core-binding factor α1 (Cbfα1), a transcription factor member of the RunX family and a key regulatory gene for osteoblasts. After RunX2 expression is stimulated, the effects of BMP/Smads and Wnt on the differentiation of cardiac valve interstitial cells into osteoblasts are focused on the upregulation of channel RunX2 expression, eventually leading to calcification. These arguments support the current conclusions. Thus, gene therapy seems promising.

### 3.2. Loading VEGF and RunX2-siRNA in the MSN and Characterization

VEGF and RunX2-siRNA were loaded into MSN (MSN@VEGF-PEI-siRNA) to control the release of VEGF and promote the intracellular property of RunX2-siRNA. The encapsulation efficiency was 17.45 ± 0.55% and 16.76 ± 0.14%, respectively. We observed that the synthesized liquid phase was white and cloudy, and the Dundar effect can be produced by light beam irradiation, indicating that the product obtained in the several key synthesis steps had good dispersion (Figure 2A). MTT assay showed that concentrations of 0.75 mg/mL and 1 mg/mL showed higher cell activity (Figure 2B). PEI modification and RunX2-siRNA complex formation did not affect the morphology and size distribution of nanoparticles but tended to slightly aggregate in solution (Figure 2C,D). This phenomenon might be due to increased particle surface charge and RunX2-siRNA surface charge crosslinking. In addition, the surface potential of nanoparticles and suitable nanoparticle size (50 nm, Figure 2I) made it easier for cells to take them up (Figure 2E,H and Appendix A). Energy dispersive spectroscopy and elemental mapping reveals the basic elemental composition of nanoparticles (Figure 2F,G). Figure 2J,K illustrates the accumulation and release curve of complex nanoparticles in vitro. VEGF will be slowly released during the first week. No initial burst was observed in the early stage, but a slow controlled release effect was achieved in the late stage. The delivery system solved the problem of the short half-life of VEGF because it remains localized at a certain concentration, avoided high dose administration, increased VEGF’s bioavailability, and avoided side effects. On the other hand, siRNA on the surface of nanoparticles showed a cascade release at 2 h that stabilized in a short time.

The cumulative release profile shows that the nano-delivery system releases only a small amount of the drug in the first few hours, and our experimental results are consistent with those in the references. The result will be that nano-delivery will release more drugs into the cells, which is a result we would like to see. When the nano-delivery system reaches the proximal nucleoplasm by endocytosis, VEGF and siRNA from the nano-delivery system are gradually released. It should be further noted that some of the VEGF may enter the nucleus. The vast majority of siRNAs are in the cytoplasm, as RNAi occurs in the cytoplasm and the presence of siRNAs in the nucleus would reduce the efficiency of silencing. This may also be one of the reasons why the silencing efficiency of MSNs loaded with siRNAs is overall higher than that of conventional transfection reagents.

It is worth mentioning that we were purposefully exploring the connection between the nano-delivery system and macrophages. The results showed that the nano-delivery system seems to increase the release of inflammatory factors by a small amount, but it was not statistically significant (Appendix A). In conclusion, based on the structural characteristics of mesoporous silica, we successfully constructed a delivery system with uniform nanoparticle size, low cell toxicity, and the ability to realize the co-delivery of VEGF and siRNA.

### 3.3. Characterize the Anti-Calcification Property of MSN@VEGF-PEI-siRNA Nanocomposite with Valvular Interstitial Cells

We constructed MSN@VEGF-PEI-siRNA nanocomposite to transfect valvular interstitial cells (VICs) and compared them with Lipo 3000, a conventional transfection reagent. The results showed that both MSN@VEGF-PEI-siRNA and Lipo 3000+siRNA groups could silence the related calcification genes, and the transfection effect of the MSN@VEGF-PEI-siRNA group was better than that of the Lipo 3000+siRNA group (Figure 3A). Alizarin red and ALP staining showed that the MSN@VEGF-PEI-siRNA group had the fewest calcified nodules, 0.163 ± 0.016 and 1.115 ± 0.095 U/mg, respectively, after quantitative analysis (Figure 3B–E). In addition, the calcium content of the MSN@VEGF-PEI-siRNA group was 16.630 ± 1.253 mg/mg, and a significant difference was detected between the MSN@VEGF-PEI-siRNA and Lipo 3000+siRNA groups (Figure 3F) (** *p* < 0.01).

A careful sequence selection and synthesis of tailored siRNAs may have enormous repercussions in therapy as almost all genes might be downregulated. Thus, this powerful approach might circumvent the limitations of traditional drug therapy, and the development of such therapeutic strategies could significantly impact modern medicine. However, the effective delivery of siRNA technology has a direct impact on whether siRNAs are able to produce their intended effects. To overcome this problem, siRNA delivery to the target region should be non-toxic and stable in order to enable the application of RNAi [30]. To date, various methods have been used (including mechanical (ultrasound), physical methods [31], electroporation [32], mouse hydrodynamic tail vein injections [33], and gene guns to deliver siRNA in vivo. In addition, local administration intraperitoneal, intravenous, subcutaneous, and chemical methods based on the synthesis of non-viral vectors (polymers [34], cationic lipids [35], and peptides [36]) have been successfully used.

Inorganic nanoparticles have emerged as alternative nanomaterials to traditional lipid formulations for siRNA delivery. With the aid of cationic lipids and polymers to form hybrid structures, MSNs have reported average gene silencing efficiencies of 75.7 ± 19.4% in vitro and 63.2 ± 17.0% in vivo [37]. The large fluctuation in transfection efficiency is due to different chemical modifications of mesoporous silica, such as PEI-MSN [38], magnetic PEI-MSN [39], PEI-MSN-KALA [40], and PEI-MSN-PEG [41,42]. PEI is one of the most studied cationic polymers for the transfection of oligonucleotides, siRNA, and plasmid DNA [43]. The high density of electric charge present in PEI is viewed as a beneficial feature for merging with siRNA, which supports the escape of the intercellular body through the proton sponge effect. The use of PEI as a guinea pig siRNA model is effective in antiviral activity in Ebola virus infection models and influenza infection mouse models [44]. The initial use of PEI to successfully transport siRNA in cancer was exhibited in a mouse model of ovarian cancer, where the inhibition of human EGF receptor 2 (HER2) was achieved [45]. The silencing efficiency of the nanocomposite was stable at 63 ± 2.1%, which was about 50% higher than that of the commonly used liposome transfection system. Therefore, we speculated that MSN@VEGF-PEI-siRNA nanocomposite has an anti-calcification effect in the valvular interstitial cell calcification model.

### 3.4. Characterize the Endothelialization Rate of the MSN@VEGF-PEI-siRNA Nanocomposite

Cell migration, tube formation, proliferation, and adhesion were measured to determine the rate at which MSN@VEGF-PEI-siRNA induces endothelialization. As shown in Figure 4A,C, MSN@VEGF-PEI-siRNA significantly enhanced endothelial cell migration compared to the other three groups. Moreover, MSN@VEGF-PEI-siRNA has an obvious capillary formation (Figure 4B). The total length of endothelial cells and the number of new tubes in each quantitative analysis group were CON < MSN < MSN-PEI-siRNA < MSN@VEGF-PEI-siRNA (Figure 4D,E). In addition, endothelial cells showed strong proliferative activity on day 3, especially in the MSN@VEGF-PEI-siRNA group. Similarly, endothelial cells in each group showed greater adhesion at 12 h, and the MSN@VEGF-PEI-siRNA group was more significant than the other groups (Figure 4F,G). These results suggested that MSN@VEGF-PEI-siRNA promotes endothelialization in vitro.

VEGF has been applied by many researchers in various fields for the modification and functionalization of biomaterials due to its ability to promote a series of physiological behaviors in endothelial cells [46,47]. The primary mechanism of VEGF is to bind to a specific receptor on the cell membrane and exert its biological effects through the VEGF/VEGFR-2 signaling pathway. Due to the short half-life of free VEGF in the body, a high dose is required if a specific concentration is required to significantly exert its biological effect. However, high doses can cause severe side effects, such as promoting the vascularization of untargeted tissues and organs and the rapid growth of some occult tumors [48]. Therefore, the best way to resolve this issue is to extend the half-life of the drugs by local drug release to reduce side effects.

### 3.5. Immobilize MSN@VEGF-PEI-siRNA Nanocomposite into DPAV and Characterization

MSN@VEGF-PEI-siRNA nanocomposite was immobilized into DPAV by polyelectrolyte layer-by-layer technology, and the representative image of scanning electron microscopy (SEM) is shown in Figure 5A. Along with the direction of the valve fiber shape, a layer of nanoparticles was coupled on the fiber surface with an irregular arrangement and slight aggregation, probably due to aggregation by electrostatic adsorption. VEGF and siRNA released from the DPAV played a crucial functional role in DPAV. As shown in Figure 5B, around 78.211 ± 1.353% of siRNA adsorbed on DPAV was released within 48 h, but stabilized at 24 h. However, the cumulative release of VEGF in MSN was about 78.062 ± 2.739%, and the results showed that the release of VEGF was stable following a sudden release behavior from 24 h to 72 h, which gradually stabilized. This is of course only the release that can be achieved in vitro, and then in vivo, it is influenced by several factors. Haemodynamics may be one of the critical determinants of whether the valve succeeds in its intended function, as the aortic valve opens and closes during a cardiac cycle, friction from the bloodstream generates shear stresses on the aortic valve fibers, and the generation of shear strains determines to a large extent whether the drug release is localized or not. Over the next 48 h, the number of cells on the valves increased in all groups, but at each time point, the DPAV-MSN@VEGF-PEI-siRNA group had the largest number of cells (Figure 5C).

We planted the interstitial cells on the modified hybrid valve and digested the cells for PCR analysis. There was downregulation of related calcification genes (RunX2, OPN, OSX, BSP, and OCN) and upregulation of IL-6 in the DPAV-MSN@VEGF-PEI-siRNA group (Figure 5D). This indicated that the modified hybrid valve could stably release siRNA and produce its biological effects. The DPAV-MSN@VEGF-PEI-siRNA group had the lowest hemolysis rate compared to the other three groups (** *p* < 0.01), and the ratio of the DPAV group was higher than the latter two groups, among which the ratio of the porcine aortic valve (PAV) group was maximal (Figure 5E). The results showed that the modified hybrid valves of the nano-delivery system could reduce hemolysis to some extent. According to Standard F 756-00 of the American Society for Materials and Testing, 2% hemolysis is considered non-hemolysis of biomaterials [49]. The modified hybrid valve did not increase hemolysis after nanoparticle modification; conversely, it improved blood compatibility. At 6 h, the number of cells attached to DPAV-MSN@VEGF-PEI-siRNA was the largest. The calcium content analysis of DPAV-MSN@VEGF-PEI-siRNA was 22.321 ± 1.103 μg/mg (Figure 5F). Significant differences were detected between the DPAV-MSN@VEGF-PEI-siRNA and DPAV-MSN@VEGF-PEI groups, the PAV and DPAV groups (** *p* < 0.01).

Although the MSN@VEGF-PEI-siRNA nanocomposite showed good anti-calcification performance, it may not be sufficient to rely on a single interfering fragment to construct a new comprehensive functional valve stent. The remaining matrix of the decellularized porcine aortic valve produces a violent antigen-antibody reaction in the body, resulting in the failure of the hybrid valve stent structure [50,51]. Previous studies have shown that valving endothelial cells prevents the extracellular matrix from producing an immune response and exerts the function of self-closed circulating blood cells in body resistance, reduces the immune response of pig aortic valve to cells, avoids the formation of platelet thrombus, and inhibits osteogenetic differentiation, thereby improving valve persistence [52].

### 3.6. Biomechanical Characterization of MSN@VEGF-PEI-siRNA Nanocomposite-Modified DPAV

To evaluate the impact of the electrostatic adsorption of nanoparticles on the mechanical properties of a hybrid valve with directional anisotropy, we conducted mechanical property tests in both circumferential and radial directions. Uniaxial tensile load tests were performed to measure the ultimate tensile strength, fracture strength, fracture strain, and elastic modulus in each group. As shown in Figure 6A, the modified hybrid valve had the shape of a common aortic valve, which was white, soft, and elastic. As shown in Figure 6B,C, the DPAV-MSN@VEGF-PEI-siRNA group and the PAV group were statistically significant for DPAV in the circumferential direction (* *p* < 0.05), while in the radial direction, no significant difference was detected between the PAV group and DPAV-MSN-PEI-siRNA for DPAV (* *p* < 0.05). The fracture strain in both directions is illustrated in Figure 6D. DPAV-MSN@VEGF-PEI-siRNA, DPAV-MSN-PEI-siRNA, and PAV showed a significant difference in the circumferential direction of DPAV (** *p* < 0.01). DPAV-MSN@VEGF-PEI-siRNA and DPAV-PEI-siRNA had significant differences in the circumferential direction of DPAV (** *p* < 0.01). In the radial direction, DPAV-MSN@VEGF-PEI-siRNA, DPAV-MSN-PEI-siRNA, and PAV had significant statistical significance for DPAV (** *p* < 0.01). As shown in Figure 6E, in the direction of the circumference, the DPAV-MSN@VEGF-PEI-siRNA group and the PAV group had statistical significance for DPAV (* *p* < 0.05). In the radial direction, a significant difference was detected between the PAV and DPAV-MSN-PEI-siRNA groups for DPAV (* *p* < 0.05). As shown in Figure 6F, the ultimate tensile strength of DPAV-MSN@VEGF-PEI-siRNA and DPAV-MSN-PEI-siRNA were not significantly different from those of PAV in the circumference and radial direction (*p* > 0.05). DPAV showed significant differences in ultimate tensile strength compared to the other three groups in the circumferential direction (* *p* < 0.05); however, there were no significant differences between DPAV and DPAV-PEI-siRNA or DPAV-MSN@VEGF-PEI-siRNA in the radial direction (** *p* < 0.01).

Currently, all decellularization methods damage the valve structure and result in a potential loss of surface composition, resulting in mechanical degradation [53]. Ultimate tensile strength, fracture strength, fracture strain, and elastic modulus are critical indexes of mechanical properties. Appropriate strength ensures the stability and durability of the specimen transplanted into the body, and appropriate modulus can ensure stable opening and closing after transplantation. Anisotropy is the key characteristic of the heart valve lobules. The high elastic modulus in the radial direction ensures rapid valve closure in the diastole, and the low elastic modulus in the circumferential direction ensures rapid valve opening in the systole.

### 3.7. Endothelialization of the MSN@VEGF-PEI-siRNA Nanocomposite-Modified DPAV In Vivo

Figure 7A shows the CD31 staining of the implanted valves. Analysis of CD31 staining (Figure 7B) further confirmed our in vitro experimental results in that there were more endothelial cells on the valve in the DPAV-MSN@VEGF-PEI-siRNA group than in the other three groups, and there were many newly formed capillaries. Because the modified hybrid valve was able to release VEGF, it aggregated more endothelial cells, thus achieving endothelialization and covering the calcification site. Further, more endothelial cell aggregation may mean faster in vivo reconstruction of the modified hybrid valve, which is an interesting result.

In summary, the results of this study highlight the potential of the modified hybrid valve in anti-calcification and endothelialization. Nevertheless, some shortcomings need to be addressed by future studies. Although we found that the modified hybrid valve can perform biological functions in vitro, in vivo experiments are needed to validate the findings. In addition, hybrid valves need to be implanted in large animal heart valve replacements to verify their function and resistance to rapid degeneration and calcification. These results strongly support the theory that the modification of the MSN@VEGF-PEI-siRNA delivery system that enables the co-delivery of VEGF and siRNA may be an effective approach for the development of decellularized functional heart valves.

## 4. Conclusions

In this study, we successfully constructed the MSN@VEGF-PEI-siRNA nanocomposite that can realize the co-delivery of VEGF and RunX-2-siRNA. We also confirmed that this nanocomposite was locally stable with sustained release and exerts anti-calcification effects. Finally, the MSN@VEGF-PEI-siRNA nanocomposite was immobilized into the decellularized porcine aortic valve by electrostatic adsorption. Thus, the modified hybrid valve had good biomechanical properties, biocompatibility, and anti-calcification properties. These promising results strongly suggest that a nanocomposite based on mesoporous silica was an effective approach to modifying the decellularized porcine aortic valve.

## Figures and Tables

**Figure 1 bioengineering-10-00859-f001:**
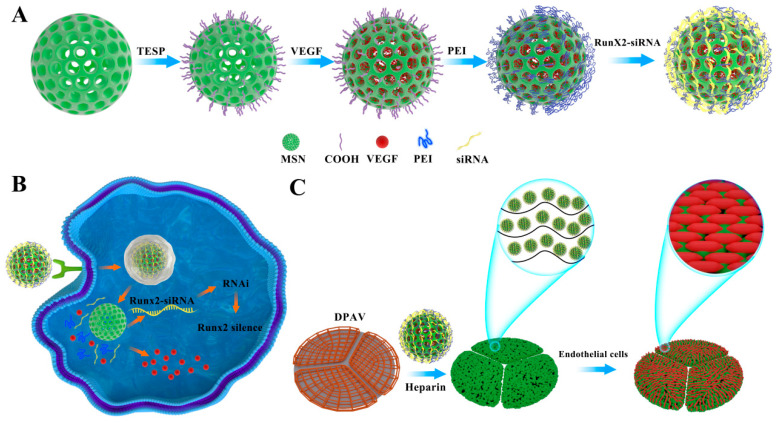
Preparation scheme. (**A**) MSN@VEGF-PEI-siRNA synthesis scheme. (**B**) Cell uptake of nanoparticles. (**C**) Preparation of modified hybrid valve.

**Figure 2 bioengineering-10-00859-f002:**
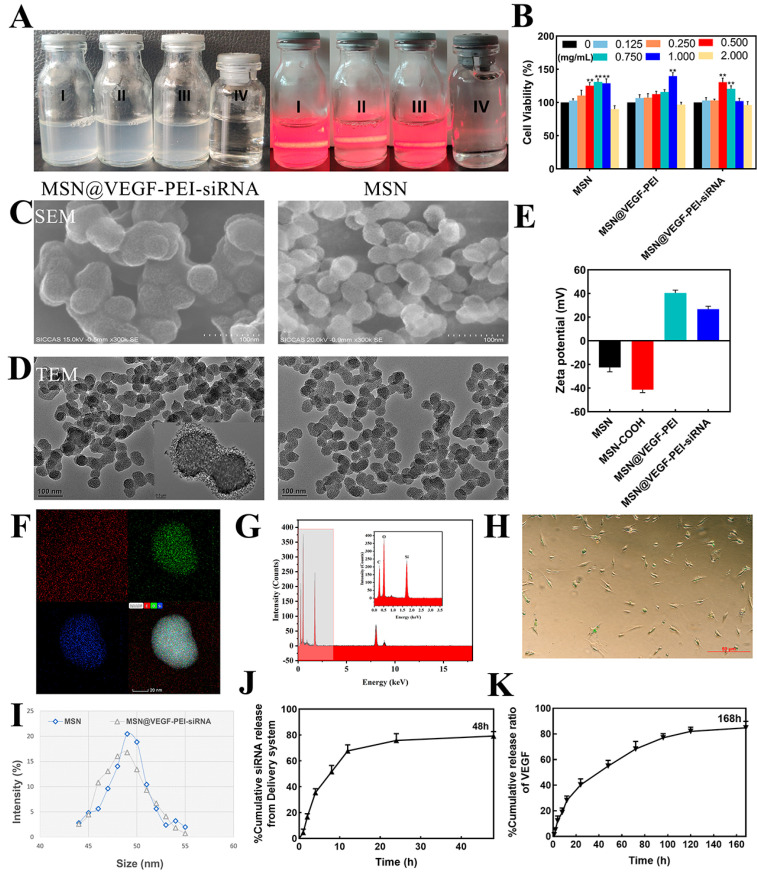
Characteristics of MSN@VEGF−PEI-siRNA delivery system. (**A**) I: MSN, II: MSN@VEGF−PEI, III: MSN@VEGF−PEI−siRNA, IV: Double steaming water. The Dundar effect was used to measure the dispersion of the synthesized material. Cytotoxicity was detected by concentration gradient in each group (MSN, MSN@VEGF−PEI, MSN@VEGF−PEI−siRNA). Concentrations of 0.75 mg/mL and 1 mg/mL showed higher cell activity. (**B**) Representative image of SEM (**C**) and TEM (**D**). Zeta potential (**E**) and elemental mapping image of MSN (**F**). Energy dispersive spectroscopy of MSN, reveals the basic elemental composition of nanoparticles (**G**). Uptake of valve interstitial cells (**H**) and particle size analysis. The nanoparticle particle size was 50 nm (**I**). Cumulative release curve of siRNA (**J**) and VEGF (**K**). Significant differences (* *p* < 0.05, ** *p* < 0.01), n = 3.

**Figure 3 bioengineering-10-00859-f003:**
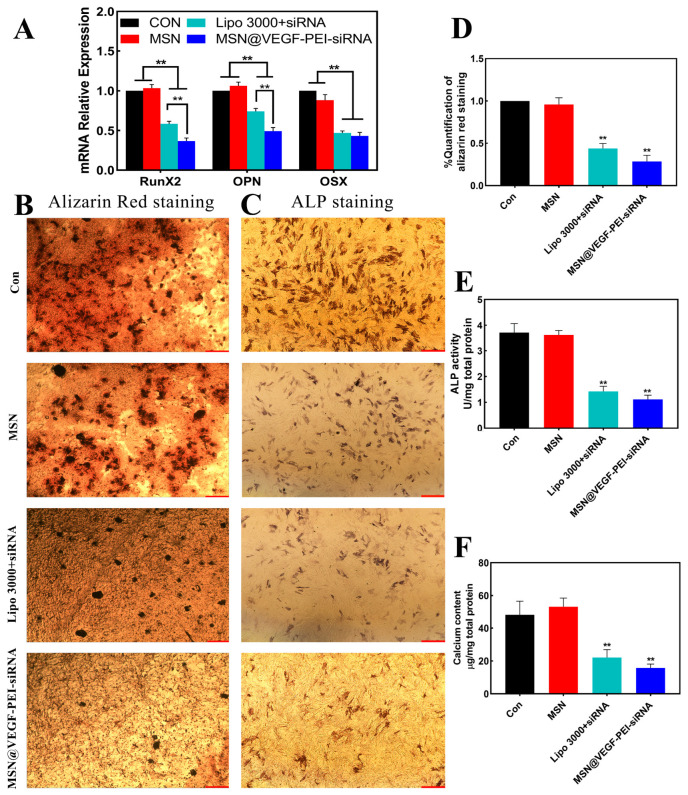
Anti-calcification of the MSN@VEGF-PEI-siRNA delivery system; MSN@VEGF-PEI-siRNA silencing efficiency is higher than that of conventional transfection reagents. mRNA expression of calcification genes associated with valve interstitial cells was detected after transfection with MSN@VEGF-PEI-siRNA and Lipo 3000+siRNA (**A**). Representative image of Alizarin red staining (**B**) and ALP staining (**C**). Quantitative analysis of alizarin red staining (**D**) and ALP staining (**E**). Determination of calcium content (**F**). Significant differences (* *p* < 0.05, ** *p* < 0.01), n = 3. scale bar = 20 µm.

**Figure 4 bioengineering-10-00859-f004:**
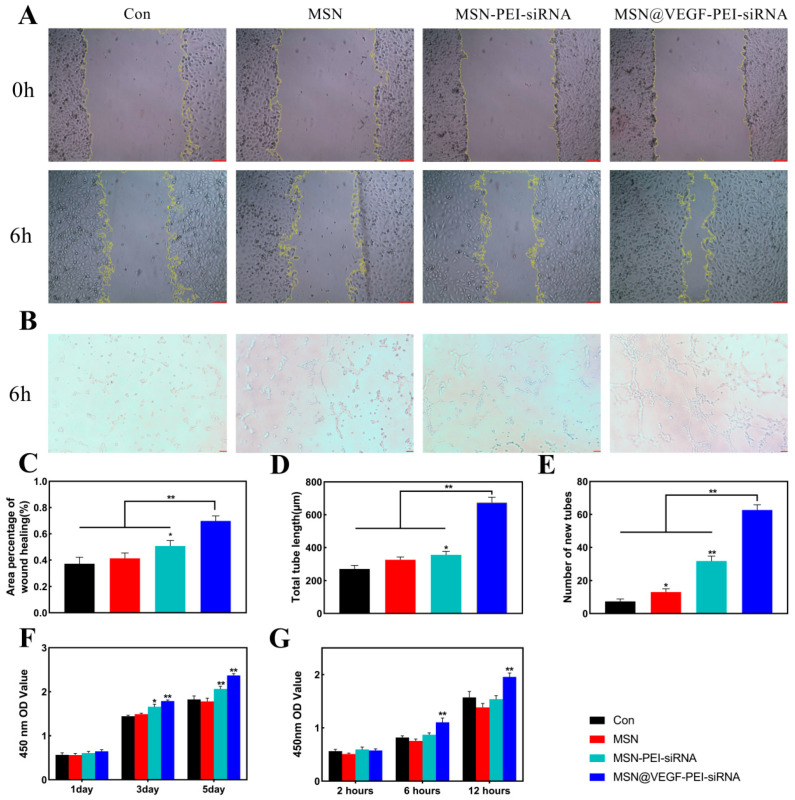
Physiological behavior of endothelial cells after treatment with MSN@VEGF-PEI-RunX2 delivery system; MSN@VEGF-PEI-RunX2 promotes endothelial cell proliferation, adhesion, migration, and tube formation. Scratch assay (**A**) and tube formation assay (**B**). The migration rate of endothelial cells (**C**). Analysis of tube formation assay—total tube length (**D**) and number of new tubes (**E**). Proliferation assay (**F**) and adhesion assay (**G**) of endothelial cells. Significant differences (* *p* < 0.05, ** *p* < 0.01), n = 3. scale bar = 50 µm.

**Figure 5 bioengineering-10-00859-f005:**
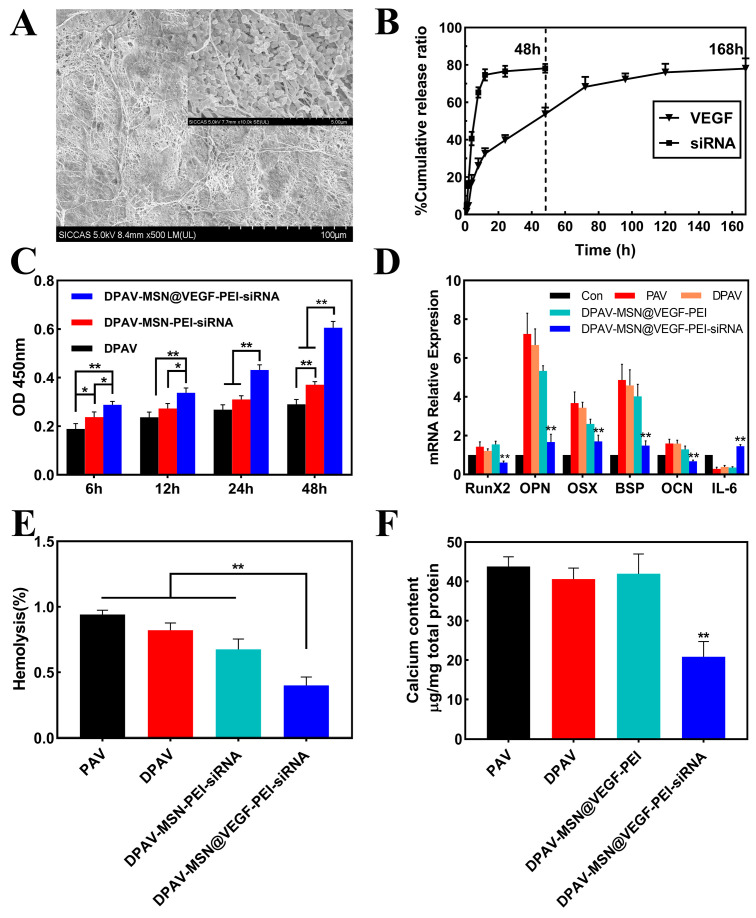
Characterization of the hybrid valve and function test after cell implantation. Representative image of the hybrid valve under a SEM (**A**). Cumulative release curve of siRNA and VEGF from hybrid valve (**B**). Proliferation and adhesion test of the hybrid valve (**C**). mRNA expression of the calcified gene was detected after implantation of valvular interstitial cells (**D**). Hemolysis rate of the hybrid valve (**E**). Calcium content was detected after the implantation of valve interstitial cells (**F**). Significant differences (* *p* < 0.05, ** *p* < 0.01), n = 3.

**Figure 6 bioengineering-10-00859-f006:**
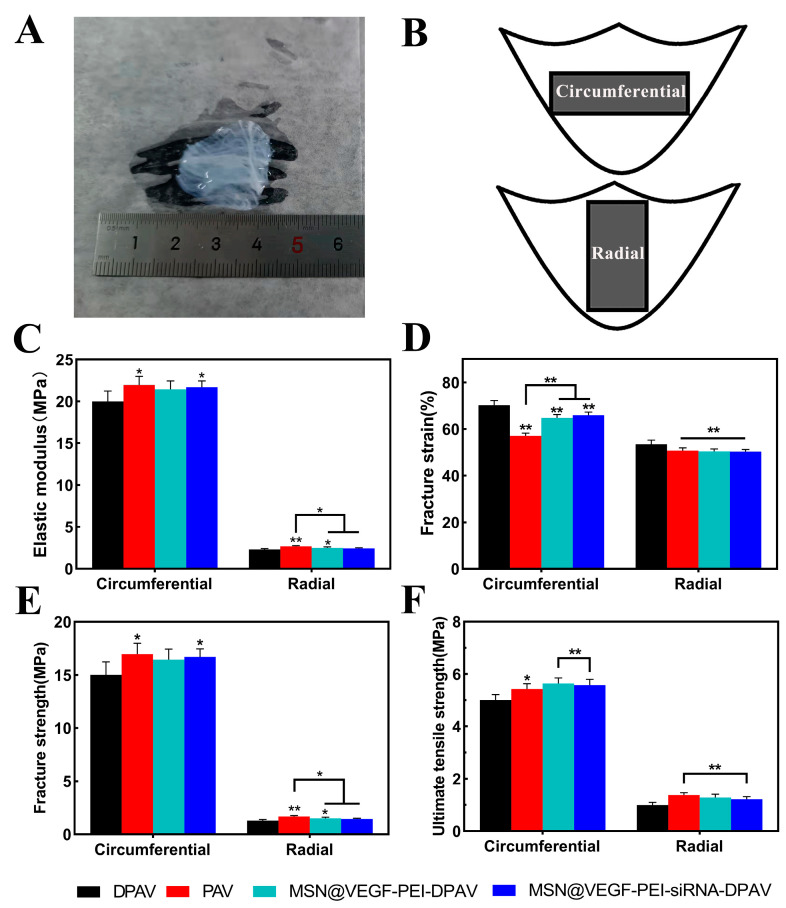
Mechanical properties of the hybrid valve. Representative image of modified hybrid valves (**A**). Diagram of circumferential direction and radial direction (**B**). Elastic modulus (**C**), fracture strain (**D**), fracture strength (**E**), and ultimate tensile strength (**F**) of the hybrid valve. Significant differences (* *p* < 0.05, ** *p* < 0.01), n = 3.

**Figure 7 bioengineering-10-00859-f007:**
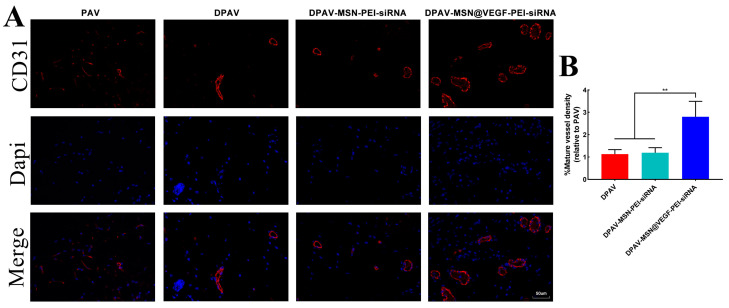
CD31 staining of PAV, DPAV, DPAV-MSN-PEI-siRNA, and DPAV-MSN@VEGF-PEI-siRNA (**A**) and analysis of CD31 staining (**B**). Significant differences (* *p* < 0.05, ** *p* < 0.01), n = 6.

## Data Availability

Not applicable.

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
