# Peer review of "Biofunctionalized Decellularized Tissue-Engineered Heart Valve with Mesoporous Silica Nanoparticles for Controlled Release of VEGF and RunX2-siRNA against Calcification"

_bioengineering, 2023, doi:10.3390/bioengineering10070859_

Round 1
Reviewer 1 Report
The authors constructed a delivery system based on mesoporous silica to biofunctionalized TEHV scaffold, with RunX2-siRNA and VEGF loaded into mesoporous silica nanoparticles and immobilized on the decellularized porcine aortic valve for endothelialization and anti-calcification. The results suggest potentials in clinical application.
- The aortic valve is constantly subjected to mechanical stimulation, and vascular cell growth is greatly influenced by mechanical force induced by hemodynamics. The in vitro model should consider effects of mechanical force upon the cultured cells or stability of the nanocomposites. Therefore, it is far from “confirmed” as stated in the Conclusion.
-The in vivo experiments with subcutaneous implantation demonstrate only representative CD31 immunofluorescence staining without reproducible data with statistical analysis. It is unclear how statistical significance was achieved, as in the legend of Figure 7. Such results failed to support the conclusion of achieving in vivo endothelialization as stated in Abstract. In addition, the animal protocol should be approved by IACUC with an approval number provided.
-
Author Response
Question 1、The aortic valve is constantly subjected to mechanical stimulation, and vascular cell growth is greatly influenced by mechanical force induced by hemodynamics. The in vitro model should consider effects of mechanical force upon the cultured cells or stability of the nanocomposites. Therefore, it is far from “confirmed” as stated in the Conclusion.
Response: First of all, thank you for your comments, and we strongly agree with you. Then the in vivo environment is complex and the occurrence of the disease is multifactorial. It is difficult for us to simulate the in vivo multi-factor environment in in vitro experiments, and it can only be verified one by one as far as possible. Of course, we will fully consider your opinion in the subsequent research.
Question 2、The in vivo experiments with subcutaneous implantation demonstrate only representative CD31 immunofluorescence staining without reproducible data with statistical analysis. It is unclear how statistical significance was achieved, as in the legend of Figure 7. Such results failed to support the conclusion of achieving in vivo endothelialization as stated in Abstract. In addition, the animal protocol should be approved by IACUC with an approval number provided.
Response: Thank you for your question. We have added the statistical analysis. In addition, we provided the approval number in the animal experiment section.
Reviewer 2 Report
The manuscript under review represents very advanced study of reducing the risk of the atherosclerotic events by decreasing calcification of the aortic valve by using the genetic therapy. The research was performed in vitro on the engineered animal tissue and based on newly constructed system of a delivery of VEGR and si-RNA by loading into vehicle of mesoporous silica nanoparticles as a nano-carrier. Authors present promising application of nanotechnology to medical biotechnology and clinical practice. The research seems to be continuation of the previous study, mentioned in Reference List, and done by some members of the team.
Introduction presents shortly the main object of the research: calcification of the aortic valve as one of high risk cardiovascular disease, current state and technical and functional problems of tissue-engineered heat valves usage for controlled endothelialization and anti-calcification. The aim of the study (lines 81-84) gives concise targets to be achieved. The last part (lines 84-90) should be transferred to the beginning of Chapter 2 as presenting general methodological parts of the research.
Chapters "Materiał and Methods" and "Results and discussion" make the largest parts of the manuscript. All materials, chemicals, bioreagents, kits, antibodies, tissue processing, cell culture are described clearly, accurately enough. Suppose, other scientists could repeat or follow similar experiments using all procedures that are presented carefully. In some points, I found abbreviations that could be difficult to recognize for scientists not familiar with them (PEG - line 63, PCL - line 65, why the control group is called DM - line122? ). Short question to this part: how determination of calcium content was performed, spectrophotometrically with alizarin? I couldn't find on-line the procedure of the kit of MingDian from Shanghai. It seems to be important as the content of calcium in VIC expressed per the total protein concentration gives crucial information on calcification of samples. Lines: 218 and 223 contain some abbreviations (Con and CCK-8) that should be explained.
Results and discussion are placed in one chapter, and in my opinion it makes the presentation and discussing of the results very clear and understandable, especially with so much experiments done. My only remark concerns the proces of pathological calcification of valve, but other aortic tissues as well. I would recommend to add or correct in lines 324- 325 the phrase "...the calcified valve has the characteristic of bone tissue.", and introduce: "...by formation of calcium phosphate solids/precipitate." It is well established that the main inorganic components of the calcified atherosclerotic plaque or advanced layer in the intima are calcium phosphates of different composition, mainly o-calcium phosphate, but not hydroxyapatite found finally in bones.
All figures in the manuscript are very helpful and of good quality, also their legends are precise.
Conclusions summarize the obtained results in very concise way, are precise, and give promises and suggestions - I found it very correct.
References are chosen properly, according to the aim of the research and performed experiments.
Author Response
Question 1、The aim of the study (lines 81-84) gives concise targets to be achieved. The last part (lines 84-90) should be transferred to the beginning of Chapter 2 as presenting general methodological parts of the research.
Response: Thank you for your comments. We have tried to change it based on your proposal. After transferring lines 84-90 to Chapter 2, we found that the original part would miss the original logic. Therefore, after the discussion of our research group, we should still maintain the original text, and the reason for doing this is that we can not only maintain the original logic but also introduce the general methodological of the research.
Question 2、Chapters "Materiał and Methods" and "Results and discussion" make the largest parts of the manuscript. All materials, chemicals, bioreagents, kits, antibodies, tissue processing, cell culture are described clearly, accurately enough. Suppose, other scientists could repeat or follow similar experiments using all procedures that are presented carefully. In some points, I found abbreviations that could be difficult to recognize for scientists not familiar with them (PEG - line 63, PCL - line 65, why the control group is called DM - line122? ).
Response: Thank you for your comments. We have made the annotations according to your request (Polyethylene glycol,PEG,line 66;Polycaprolactone,PCL,line 67). The MD group serves as a control group in the in vitro calcification model, as the OIM group is necessary for treatment.
Question 3、Short question to this part: how determination of calcium content was performed, spectrophotometrically with alizarin? I couldn't find on-line the procedure of the kit of MingDian from Shanghai. It seems to be important as the content of calcium in VIC expressed per the total protein concentration gives crucial information on calcification of samples. Lines: 218 and 223 contain some abbreviations (Con and CCK-8) that should be explained.
Response: Thank you for your comments, We have added the protocols to the manuscript. The protocols are as follows:1ml 10% CPC was Added to each well of the six-well plate. Then, Gently shake on a shaker for 20 minutes and transfer the eluted CPC into an EP tube. Dilute the eluted CPC 20 times with 10% CPC and measure at a wavelength of 562nm. For the detection method of calcium content, the protocols are as follows: Remove the culture medium and wash once with PBS solution. 100µl of sample lysis buffer was added to fully lyse the cells. After thorough lysis, centrifuge at 4℃ for 10,000g for 5 minutes and collect the supernatant. Measure the absorbance at 575nm. We have annotated CCK-8 in the Materials section and annotated Con in line 236.
Question 4、My only remark concerns the proces of pathological calcification of valve, but other aortic tissues as well. I would recommend to add or correct in lines 324- 325 the phrase "...the calcified valve has the characteristic of bone tissue.", and introduce: "...by formation of calcium phosphate solids/precipitate." It is well established that the main inorganic components of the calcified atherosclerotic plaque or advanced layer in the intima are calcium phosphates of different composition, mainly o-calcium phosphate, but not hydroxyapatite found finally in bones.
Response: Thanks for your suggestions, we have revised some phrase according to your suggestions (line 345-348).
Reviewer 3 Report
Major issues concerning the overall study concept should be properly addressed.
- The loading efficiency of both VEGF and siRNA was also not discussed. It’s doubtful that all reactions would lead to 100% efficiency. In addition, how does loading the VEGF affect the amount of siRNA?
- Is the addition of VEGF crucial to the attachment of the siRNA? If not, then why was there no test sample only containing the siRNA?
- Why did the authors stop the release measurement at 80%? What was the concentration basis of the 100% release of both VEGF and siRNA?
- In relation to loading concentrations, how much MSN is able to be loaded into the DPAV?
- It is also doubtful that the attachment of MSN@VEGF-PEI-siRNA did not affect the release profile of VEGF and siRNA.
- The authors also did not discuss the release mechanism of both VEGF and siRNA loaded into the MSN. This is an important detail with regard to the suitability of this material for preventing valve calcification.
Although the core concept of the manuscript is interesting, the presentation and writing need substantial editing by a scientific English writer. Several statements are quite confusing and reduce their overall readability.
- Sections 2.1 and 2.2 should be combined into 1 since they are only related to the isolation, characterization, and osteogenic induction of VIC. It should be noted that
- Please specify the source of endothelial cells.
- Please specify the Elisa kit used for measuring inflammatory factors from the macrophage experiment.
- The word “planting” isn’t really used in the context of in vitro experiments. It should be changed into “seeding”. In addition, this section is highly confusing since it included two experiments- the cell viability and hemolysis test. This section should be written separately and described in more detail. Was the hemolysis test performed on cell-seeded samples?
- Please modify the caption for all the figures. Aside from providing the annotations, Summarize the major findings based on the gathered data. This will drastically improve the readability of the manuscript.
- Rather than presenting the raw O.D. values, please convert them to normalized data and use either fold change or percentage (%).
- Figure 6. B doesn’t really provide a good representation of what the authors want to describe.
- In section 2.10, please provide the description of each experimental group.
- The amount of MSN@VEGF-siRNA loaded unto the DPAV is unclear. How does the process of loading the nanoparticles affect the total amount of VEGF and siRNA in the resulting complex (DPAV-MSN@VEGF-PEI-siRNA)? Is there any loss of VEGF and siRNA? If not, how is it possible? Additional supplementary data should be provided regarding this detail is needed.
- What do the authors mean by “calcified media”?
Author Response
Major issues concerning the overall study concept should be properly addressed.
Question 1、The loading efficiency of both VEGF and siRNA was also not discussed. It’s doubtful that all reactions would lead to 100% efficiency. In addition, how does loading the VEGF affect the amount of siRNA?
Response: Thank you for your suggestions. The loading process and calculation method of loading rate were added and described at Results (line 359). On the question of whether VEGF affects the amount of siRNA, our design concept is that they do not interfere with each other. VE GF is loaded into the MSN by impregnation, while siRNA is fixed on the surface of MSN by electrostatic adsorption. Our results also support the above view. We therefore suggest that both the loading of VEGF and siRNA and their release do not interfere with each other.
Question 2、Is the addition of VEGF crucial to the attachment of the siRNA? If not, then why was there no test sample only containing the siRNA?
Response: Thanks for your valuable comments, VEGF does not affect siRNA attachment. Of course, the suggestions you proposed is the best logical, and only sample containing siRNA can be excluded after testing. However, in terms of the function of the material, the functions of VEGF and siRNA produced in the in vitro tests are different, and the experimental grouping is also able to show that the siRNA plays our expected role. It is difficult to verify that VEGF functions together with siRNA in in vitro tests. As mentioned at the end of the manuscript, further in vivo experiments are needed to verify whether VEGF and siRNA play a synergistic role or a separate role in vivo.
Question 3、Why did the authors stop the release measurement at 80%? What was the concentration basis of the 100% release of both VEGF and siRNA?
Response: Thank you for your suggestions. Instead of stopping measurement at 80%, we set the time point for measurement. In terms of VE GF release, we tested VEGF release at 336h and found barely detectable presence of VEGF during the one-week interval. We therefore believe that VE GF release in the nanoparticles before 168h or 168h has reached the limit. 100% release drug volume we calculated from the drug load per mg of nanoparticles. In theory, it is difficult to achieve a 100% release rate because of the inevitable loss of some drugs during the synthesis process, or when the nanoparticles are not released.
Question 4、In relation to loading concentrations, how much MSN is able to be loaded into the DPAV?
Response: Thank you for your comments. Firstly the loading of MSN can be calculated based on the total amount of MSN minus the amount of solution remaining after loading. Its loading can be related to the researcher's design, and the electrolyte layer-by-layer technique can be loaded with different layers of nanoparticles
Question 5、It is also doubtful that the attachment of MSN@VEGF-PEI-siRNA did not affect the release profile of VEGF and siRNA.
Response: Thank you for your comments. The attachment of MSN@VEGF-PEI-siRNA is only by electrostatic adsorption and not by modification which closes the pores of MSN and is therefore unaffected. Our previous study [1], although utilising different nanoparticles immobilised on DPAV by covalent modification, did not have a large impact on release, despite the stronger effect of this method.
Question 6、The authors also did not discuss the release mechanism of both VEGF and siRNA loaded into the MSN. This is an important detail with regard to the suitability of this material for preventing valve calcification.
Response: Thank you very much for your suggestions, which have helped us a lot with the manuscript. We have added an elaboration of the mechanism to the manuscript (line 378-388 ). Of course we did not perform experiments involving part of the release mechanism, so we have quoted from a very informative reference [2]. The cumulative release profile shows that the nano-delivery system releases only a small amount of the drug in the first few hours, and our experimental results are consistent with those in the references. The result will be that nano-delivery will release more drugs into the cells, which is a result we would like to see. When the nano-delivery system reaches the proximal nucleoplasm by endocytosis, VEGF and siRNA from the nano-delivery system are gradually released. It should be further noted that some of the VEGF may enter the nucleus. The vast majority of siRNAs are in the cytoplasm, as RNAi occurs in the cytoplasm and the presence of siRNAs in the nucleus would reduce the efficiency of silencing. This may also be one of the reasons why the silencing efficiency of MSNs loaded with siRNAs is overall higher than that of conventional transfection reagents. Of course, PEI also plays an important role in the nano-delivery system, and PEI modification could better enable the nano-delivery system to escape from the lysosome and protect the siRNA from degradation. All of the above points support the conclusions we have drawn.
Comments on the Quality of English Language
Question 7、Sections 2.1 and 2.2 should be combined into 1 since they are only related to the isolation, characterization, and osteogenic induction of VIC. It should be noted that
Response: We have combined Section 2.1 and 2.2.
Question 8、Please specify the source of endothelial cells.
Response: We have added the endothelial cell source to the manuscript (line 120).
Question 9、Please specify the Elisa kit used for measuring inflammatory factors from the macrophage experiment.
Response: We have added the source of the kit for testing the inflammatory factors in the manuscript (line 106).
Question 10、The word “planting” isn’t really used in the context of in vitro experiments. It should be changed into “seeding”. In addition, this section is highly confusing since it included two experiments- the cell viability and hemolysis test. This section should be written separately and described in more detail. Was the hemolysis test performed on cell-seeded samples?
Response: Thank you for your valuable comments. We have changed the word "planting" to "seeding". In accordance with your comments, we have divided the cell seeding and haemolysis tests into two parts, which greatly avoids any possible confusion on the part of the reader, as can be seen in detail in sections 2.12 and 2.13 of the manuscript.
Question 11、Please modify the caption for all the figures. Aside from providing the annotations, Summarize the major findings based on the gathered data. This will drastically improve the readability of the manuscript.
Response: Thank you for your comments. Following your comments, we have changed the caption of all the figures. We have also made a brief summary based on the data, which we hope will be to your satisfaction.
Question 12、Rather than presenting the raw O.D. values, please convert them to normalized data and use either fold change or percentage (%).
Response: In accordance with your comments, we have converted the relevant data.
Question 13、Figure 6. B doesn’t really provide a good representation of what the authors want to describe.
Response: Figure 6B shows a schematic representation of the mechanical properties of the valve as we test them in the circumferential and radial directions. This is an important concept throughout Figure 6 and is therefore highlighted in line 531.
Question 14、In section 2.10, please provide the description of each experimental group.
Response: We have supplemented this with a description of the experimental group in section 2.9 (Line 236).
Question 15、The amount of MSN@VEGF-siRNA loaded unto the DPAV is unclear. How does the process of loading the nanoparticles affect the total amount of VEGF and siRNA in the resulting complex (DPAV-MSN@VEGF-PEI-siRNA)? Is there any loss of VEGF and siRNA? If not, how is it possible? Additional supplementary data should be provided regarding this detail is needed.
Response: Thank you for your comments. As described in the manuscript, the amount of MSN@VEGF-PEI-siRNA loaded can be calculated based on the total amount of MSN@VEGF-PEI-siRNA minus the amount remaining in the solution after loading. Whether the amount of VEGF and siRNA is lost during the loading of the nanoparticles, unfortunately, this we cannot determine because, in subsequent release assays, DPAV-MSN@VEGF-PEI-siRNA was unable to achieve 100% release, so we cannot tell whether the missing VEGF is still within the nanoparticles or was lost during loading. However, it is undeniable that some VEGF and siRNA are inevitably lost during the loading process. Still, from the data tested, the VEGF and siRNA released locally from the valves produced their respective biological effects.
Question 16、What do the authors mean by “calcified media”?
Response: Thank you for your suggestion. This was a mistake we made in the description. We have changed the word "calcified media" to "OIM".
Round 2
Reviewer 1 Report
- Cardiac valves are constantly subjected to mechanical forces generated in a beating heart. It is hard to mimic the physiological milieu using static culture, but some devices allow the study of cell/tissue under such conditions. The authors cannot claim or “confirm” (line 589) that the stability of nanocomposites, release kinetics, or anti-calcification effect may occur in situ. They should discuss the factors that may hinder reproducing the effects after valve replacement in vivo.
- The authors added a bar figure in Figure 7 with the measurement of mature vessel density in the revised version. However, there was no description of the added data in the text of Methods or Results. Can vessel density represent endothelialization of the valve?
- Figure 5 “E” is missing. The marks in Figure 7 should be consistent with those in the figure legend.
the language can be improved.
Author Response
Q1、Cardiac valves are constantly subjected to mechanical forces generated in a beating heart. It is hard to mimic the physiological milieu using static culture, but some devices allow the study of cell/tissue under such conditions. The authors cannot claim or “confirm” (line 589) that the stability of nanocomposites, release kinetics, or anti-calcification effect may occur in situ. They should discuss the factors that may hinder reproducing the effects after valve replacement in vivo.
Response: Thank you for your comments. First of all, our group very much recognizes your point of view and your proposal will make the manuscript more convincing. However, I am sorry to say that our lab does not have the equipment that can simulate the in vivo environment for cell/tissue study at this moment. On the other hand, some of our experimental protocols are also referenced from some references[1]. Of course, your comments have made our group aware of the shortcomings, and we will take them into full consideration in our future studies, thank you for your help. In addition, we have discussed in the results section (lines 474-480) whether local release can be achieved.
Q2、The authors added a bar figure in Figure 7 with the measurement of mature vessel density in the revised version. However, there was no description of the added data in the text of Methods or Results. Can vessel density represent endothelialization of the valve?
Response: Thank you for your comments. We have described the data for the CD31 staining analysis in the Results section of the revised version. CD31 is currently the single best marker of endothelial differentiation, and the more positive cells that appear with CD31 staining, the more endothelial cells are demonstrated to aggregate, resulting in the formation of ring-like microvessels or mature blood vessels. The concept of endothelialization is the formation of a membranous tissue of endothelial cells on the surface of the implant.
Q3、Figure 5 “E” is missing. The marks in Figure 7 should be consistent with those in the figure legend.
Response: Thanks to your comments, we have modified Figure 5. and we have described it in the figure legend.
[1]Wang Y, Ma B, Liu K, Luo R, Wang Y. A multi-in-one strategy with glucose-triggered long-term antithrombogenicity and sequentially enhanced endothelialization for biological valve leaflets. Biomaterials. 2021;275:120981. doi: 10.1016/j.biomaterials.
Reviewer 3 Report
The authors have satisfactorily addressed the issues raised in the previous review—no further comments.
The authors have satisfactorily addressed the issues raised in the previous review—no further comments.
Author Response
Thank you for your comments, we have changed some of the language descriptions in the revised version!